# HIERARCHICAL GRAPH LEARNERS
# FOR CARDINALITY ESTIMATION

## ABSTRACT

Cardinality estimation – the task of estimating the number of records that a database query will return – is core to performance optimization in modern database systems. Traditional optimizers used in commercial systems use heuristics that can lead to large errors. Recently, neural network based models have been proposed that outperform the traditional optimizers. These neural network based estimators perform well if they are trained with large amounts of query samples. In this work, we observe that data warehouse workloads contain highly repetitive queries, and propose a hierarchy of localized on-line models to target these repetitive queries. At the core, these models use an extension of Merkle-Trees to hash query graphs which are directed acyclic graphs. The hash values can divisively partition a large set of graphs into many sets, each containing few (whole) graphs. We learn an online model for each partition of the hierarchy. No upfront training is needed; on-line models learn as the queries are executed. When a new query comes, we check the partitions it is hashed to and if no such local model was sufficiently confident along the hierarchy, we fall-back onto a default model at the root. Our experimental results show that not only our hierarchical on-line models perform better than the traditional optimizers, they also outperform neural models, with robust errors rates at the tail.

## 1 INTRODUCTION

Cardinality estimation plays a pivotal role in query optimization of relational databases, as the query optimizer uses these estimates to order the operators in the query graph and minimize data movement. The goal of cardinality estimation is to estimate the number of records returned by each query operator to answer a SQL query, without actually executing the query. Traditional cardinality estimation methods in databases like PostgreSQL rely on single column statistics (e.g., histogram and sketches), sampling, and sometimes "magic" constants. These methods, however, can lead to significant estimation errors when underlying data assumptions, such as independence between table column and uniform data distribution within columns, are violated (Leis et al., 2015).

Recently, several methods propose neural models for cardinality estimation (Kipf et al., 2019; Zhu et al., 2021; Negi et al., 2023), without making such simplifying assumptions. The core idea frames cardinality estimation as supervised learning and train machine learning models on representative (query, cardinality) observations. While learned methods show promising results, they require a large number of training data. Note that running lots of queries, especially over large collections of data, to collect training labels is very expensive, probably requiring hours-to-days of human and machine time.

We observe that database workloads in cloud databases for analytical workloads such as Google BigQuery or Amazon Redshift contain highly repetitive queries (van Renen et al., 2024) - 50% of the real world clusters have more than 90% queries repeated in templates (only changing the constant parameters). In this paper, we focus on these workloads and propose a hierarchy of localized on-line models to target these repetitive queries. Our method falls back to a default model for non-repetitive queries. These models use an extension of Merkle-Trees to hash query graphs which are directed acyclic graphs. The hash values can divisively partition a large set of graphs into many sets, each containing few (whole) graphs. We learn a separate model for each partition of the hierarchy. While

graph sizes can vary, graphs of identical structure (within a partition) must all have the same total feature dimensionality.

Briefly, to enable this, our method employs *templatizers*. Each templatizer removes features $\mathbf{X}$ from the input graph $G$ emitting remaining graph structure ("template") $T$. We then compute a hash $\#_T$ of the template $T$. A canonical and permutation-invariant ordering of nodes, preserves their position within the feature vector. For inference on test query, cardinality is estimated using all $\mathbf{X}$'s sharing the same hash of the test graph. We start searching the hierarchy at the leaves; if the current template has enough data points to make a prediction, then we use the model at that level, otherwise to move to the next level, until we fall back to a default model at the root, which can be a traditional optimizer or a learned cardinality estimator.

Our experimental studies show that our model can already learn to predict cardinality with a high accuracy especially if repetitiveness is high. Our models outperform traditional and neural models, and produce better accuracy even at the tail (P90 and P95). Moreover, by organizing the templates in a hierarchy, we show that we can learn robust models since leaf templates are more specific and thus can be trained with a few examples while templates in the higher levels need more examples but are better in generalizing in case queries are different from what we have seen so far.

**Outline.** The rest of the paper is organized as follows: In §2, we define hierarchical graph templates and discuss the core method. §3 describes how these hierarchical graph learners are used for cardinality estimation. §4 and §5 contain the detailed experimental study and the related work, respectively. Finally, we conclude in §6.

## 2 HIERARCHICAL GRAPH TEMPLATES

### 2.1 DEFINITIONS

**Basic Notation.** Let $[n]$ be the set of integers $\{1, 2, \ldots, n\}$. Let $\pi \in \mathbb{Z}^n$ be a permutation of $[n]$. Let $\{0, 1\}^h$ be a bit-vector of length $h$ and let $\{0, 1\}^*$ denote a bit-vector of arbitrary length. We denote a (cryptographic) hashing function $\$ : \{0, 1\}^* \to \{0, 1\}^h$. Functional $\text{dom}(.)$ accepts a function as an argument and returns the domain of the argument.

**Heterogeneous Directed Acyclic Graphs.** Let $\mathcal{G}$ denote the space of heterogeneous directed acyclic graphs (DAGs). An instance $G \in \mathcal{G}$ has three parts: $G = (\mathcal{V}, \mathcal{E}, f)$, respectively, (nodes, edges, features). Let $|\mathcal{V}|$ denote the cardinality of $\mathcal{V}$. For simplicity, we assume nodes are integers, *i.e.*, $\mathcal{V} \triangleq [|\mathcal{V}|]$. We assume edge-set $\mathcal{E} \subset \mathcal{V} \times \mathcal{V}$ encodes a DAG. This assumption is **necessary** for our DAG hashing function (§2.4). Finally, every node $v \in \mathcal{V}$ has an associated "*feature dictionary*" $f^{(v)}$. We demonstrate two example $f^{(v)}$'s (pertaining to our application, §3):

$$f^{(v)} = \{\text{n: "movies", c: 10000, i: 5days }\}, \qquad \text{for } v = \boxed{\text{movies}} \text{ table in Fig. 2a; (1)}$$

$$f^{(v)} = \{\text{n: "year", t: int, u: 65, min: 1960, max: 2024 }\}, \quad \text{for } v = \boxed{\text{year}} \text{ column in Fig. 2a (2)}$$

$f^{(v)} : \mathbb{Z} \to \Psi$ can be interpreted as a function that maps categories ($\in \mathbb{Z}$) onto arbitrary objects ($\in \Psi$). Our algorithm handles any object types, however, objects (1) must be representable as $\{0, 1\}^*$ (see §2.4) and (2) if it is used for learning, must be accompanied with **featurizer** function $\psi : \Psi \to \mathbb{R}^{d_\psi}$, where $d_\psi \in \mathbb{Z}_+$ is dimensionality of extracted feature (see §3.2). We use subscript notation to access feature values: $f_u^{(v)}$ denotes the value at key $u$ (in Eq. 2, $f_u^{(v)} = 65$). Notation $f_{\mathbf{S}}^{(v)}$ reads a set of features. Formally,

$$f_S^{(v)} = \{\text{s: } f_{\text{s}}^{(v)} \mid s \in \mathbf{S}\} \quad \text{for all} \quad \mathbf{S} \subseteq \text{dom}(f^{(v)}). \tag{3}$$

For instance, $f_{\mathbf{S}}^{(v)} = \{\text{c: 10000, i: 5days }\}$ when $\mathbf{S} = \{\text{c, i}\}$ and $f^{(v)}$ is defined per Eq. 1.

**Definition 1 (GRAPH ISOMORPHISM)** $G_1 = (\mathcal{V}, \mathcal{E}_1, f)$ *is isomorphic to* $G_2 = (\mathcal{V}, \mathcal{E}_2, z)$, *denoted as* $G_1 \cong G_2$ *(equivalently, $G_2 \cong G_1$), if-and-only-if there exists a permutation $\pi$ such that* $\mathcal{E}_1 = \{(\pi_u, \pi_v) \mid (u, v) \in \mathcal{E}_2\}$ *and* $f^{(v)} = f^{(\pi_v)}$ *for all $u \in \mathcal{V}$.*

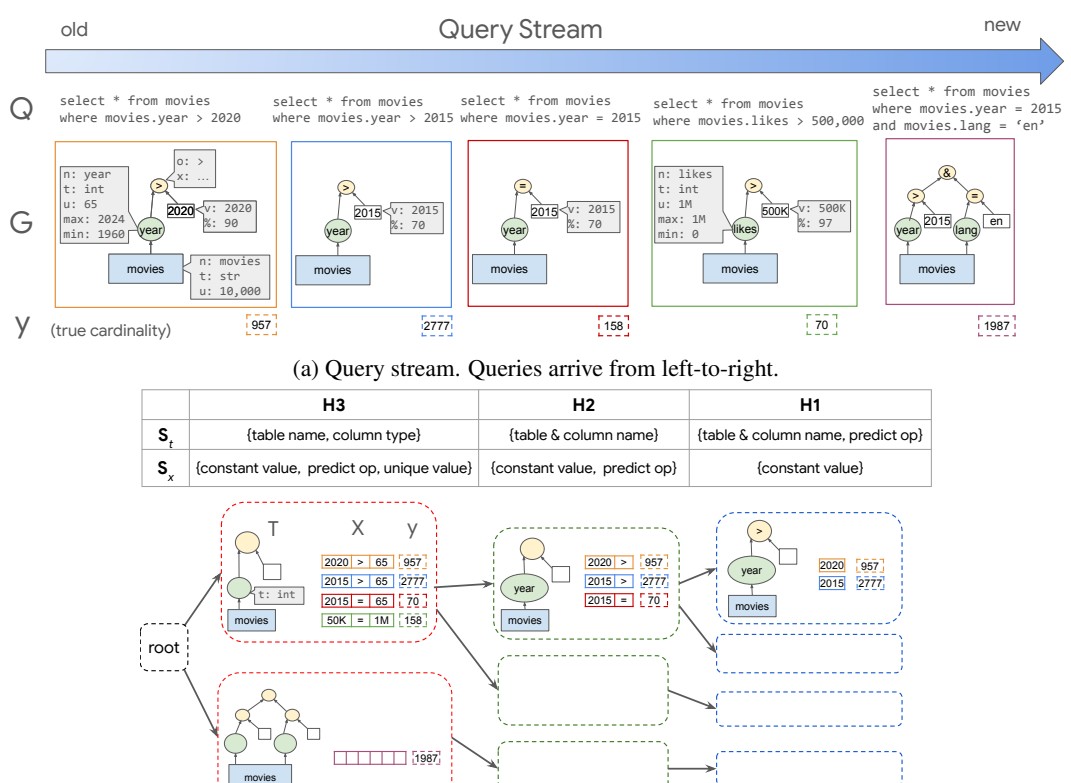

(a) Query stream. Queries arrive from left-to-right.

(b) **Template Hierarchy**. Columns correspond to template functions $H_i \in \mathcal{H}$, with Feature-Label $(\mathbf{X}_i, \mathbf{y}_i)$ per template. Leaf templatizer $H_1$ is the most-granular, grouping identical graphs with constant feature removed. Inference invokes models within each group, along one path from root to leaf (determined by $\mathbf{T}$).

Figure 1: Stream of query graphs get indexed into the template hierarchy. Every graph will store its features on a **path from root-to-leaf**. Border-colors of stream queries correspond to $(\mathbf{X}, \mathbf{y})$ pairs.

**Definition 2 (TOPOLOGICAL ORDER)** *For any directed acyclic graph $G = (\mathcal{V}, \mathcal{E}, f)$, there exists one-or-more valid topological orderings. Let $\pi \in \mathbb{Z}^{|\mathcal{V}|}$ denote one valid ordering. $\pi$ is considered a valid ordering if $\pi_v < \pi_{v'}$ for all $(v, v') \in \mathcal{E}$.*

Definition 2 implies that $v$ should be ordered before $v'$ for all edges $v \rightarrow v'$. However, it is important to remember that topological order is not unique. DAGs can have many valid topological orderings.

## 2.2 TASK: ONLINE SUPERVISED LEARNING ON GRAPHS

Our task falls under *supervised learning **on** graphs* (not *within*[1] graphs). For each graph $G \in \mathcal{G}$, we can obtain its (ground-truth) training label as $y(G) \in \mathcal{Y}$. We are interested in model $\widehat{y} : \mathcal{G} \rightarrow \mathcal{Y}$ to approximate $y(G)$ for every $G \in \mathcal{G}$. Graph Neural Networks (GNNs) (Chami et al., 2022), with graph-pooling, are valid candidates for $\widehat{y}$.

Further, we are interested in an **online setting**. Databases can receive query stream from users, during which, cardinality estimates can be obtained per incoming query (*e.g.*, to optimize join-order). We wish to **incrementally improve our models**, as we collect observations from the stream.

## 2.3 GRAPH TEMPLATE EXTRACTIONS

We define "***templatizer***" function $H : \mathcal{G} \rightarrow \mathcal{G} \times \mathbb{R}^d$. Given graph $G \in \mathcal{G}$, The outputs of $H(G)$ are (1) "*template*" $T \in \mathcal{G}$, *i.e.*, copy of the graph structure of $G$ but many features are removed and (2)

---

[1]While many recent GNN methods focus on node- or edge-level tasks, *e.g.*, node-classification or link-prediction, our method is designed for graph-level tasks, *e.g.*, graph classification or regression.

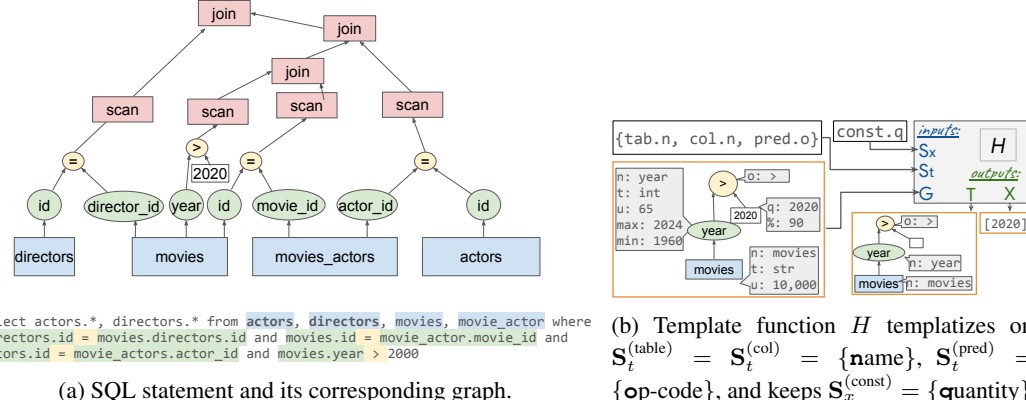

(a) SQL statement and its corresponding graph.

```
select actors.*, directors.* from actors, directors, movies, movie_actor where
directors.id = movies.directors.id and movies.id = movie_actor.movie_id and
actors.id = movie_actors.actor_id and movies.year > 2000
```

(b) Template function $H$ templatizes on $\mathbf{S}_t^{(\text{table})} = \mathbf{S}_t^{(\text{col})} = \{\mathbf{n}\text{ame}\}$, $\mathbf{S}_t^{(\text{pred})} = \{\mathbf{o}\text{p-code}\}$, and keeps $\mathbf{S}_x^{(\text{const})} = \{\mathbf{q}\text{uantity}\}$

Figure 2: Query Graph. Features shown in zoom around **year** node, depicting templatization (§2.3).

$\mathbf{x} \in \mathbb{R}^{d_T}$ the removed features. Note: $T$ determines the dimension of $\mathbf{x}$.

$$(T, \mathbf{x}) \leftarrow H(G), \quad \text{with} \quad \text{graph "\textbf{\textit{template}}" } T \quad \text{and} \quad \text{"\textbf{\textit{specialization values}}" } \mathbf{x}. \quad (4)$$

Importantly, $T$ has the ***same structure*** as $G$. However, $T$ is likely to miss many node features of $G$. Instead those features are folded onto $\mathbf{x}$.

Now, suppose two graphs, *e.g.*, $G_1 \in \mathcal{G}$ and $G_2 \in \mathcal{G}$, (i) share the same structure but (ii) have different feature values. Given an $H$, let $(T_1, \mathbf{x}_1) \leftarrow H(G_1)$ and let $(T_2, \mathbf{x}_2) \leftarrow H(G_2)$. We desire $H$ such that $T_1 \cong T_2$ due to (i) and that $\mathbf{x}_1 \neq \mathbf{x}_2$ due to (ii).

We write-down general form for all $H \in \mathcal{H}$. Specifically, each $H$ has the following form, though different $H$'s only differ in their hyperparameters.

$$(T, \mathbf{x}) \leftarrow H(G; \{\mathbf{S}_t^{(v)}, \mathbf{S}_x^{(v)}\}); \text{ with } \textbf{input } G = (\mathcal{V}, \mathcal{E}, f), \textbf{ hyperparameters } \mathbf{S}_t^{(v)}, \mathbf{S}_x^{(v)} \subseteq \text{dom}(f^{(v)}),$$
$$\text{and } \textbf{outputs } T = (\mathcal{V}, \mathcal{E}, f_{\mathbf{S}_t}) \text{ and } \mathbf{x} = \underset{v \in \mathcal{V}, s \in \mathbf{S}_x^{(v)}}{\text{CONCAT}} (\psi(f_s^{(v)})). \quad (5)$$

In other words, the output template graph $T$ keeps only features of $G$ that are listed in (hyper-parameters) $\{\mathbf{S}_t^{(v)}\}_{v \in \mathcal{V}}$, and the output features $\mathbf{x}$ is a concatenation of node features specified in (hyperparameters) $\{\mathbf{S}_x^{(v)}\}_{v \in \mathcal{V}}$. Figure 2b depicts an example $H$. Further, §2.5 utilizes family of templatizers $\mathcal{H} = \{H_1, H_2, \dots\}$. We discuss the design of $\mathcal{H}$ in §3.1.

## 2.4 One-way Hashing of Directed Acyclic Graphs

Hashing functions $\$ : \{0,1\}^* \to \{0,1\}^h$ convert a bit-vector of arbitrary length into a hash value: a fixed-size bit-vector. In our work, we desire a function that can hash DAGs, specifically, we desire:

$$\# : \mathcal{G} \to \{0,1\}^h, \quad \text{and denote} \quad \#_G \triangleq \#(G), \quad (6)$$

such that, if $\#_{G_1} = \#_{G_2}$, then $G_1 \cong G_2$ with high probability.

We design graph hash function $\#$ by generalizing the celebrated Merkle Trees (Merkle, 1988)—well-established in cryptography and computer security—onto DAGs (represneting query graphs). Merkle Trees can verify if a large file (with $n$ blocks) has been tampered with, and if so, can (efficiently) determine which block has been modified (in $\mathcal{O}(\log n)$ time). While Merkle Trees satisfy its intended use-cases, it does not naively operate on arbitrary DAGs where all nodes have features. We propose a generalization onto (i.) DAGs, (ii.) where all node may have features and (iii.) order of children is irrelevant (for most nodes). Table 6 (Appendix) summarizes the generalization.

The algorithm is relatively simple: locally hash the features in all nodes. Then, in topological order, update every node's hash to incorporate the hash of its predecessors. Finally, combine all hashes according to topological order, breaking ties using hash values. Algorithm. 2 is listed in Appendix.

---

**Algorithm 1** Procedures of Template History Learner

---

1: **input hyperparameter:** $\mathcal{H} = \{H_1, H_2, \dots\}$ (defined in Eq. 5)
2: **initialize:** $\mathcal{F} \leftarrow \{\}$
3: **initialize:** $\widehat{y} \leftarrow \text{MASTERMODEL}()$ per Eq.10
4: **function** ADDEXAMPLE$(G, y(G))$
5:      **for** $H_i \in \mathcal{H}$ **do**
6:          $(T, \mathbf{x}) \leftarrow H_i(G)$
7:          $\mathbf{X}_i^{[\#T]} \leftarrow \mathbf{X}_i^{[\#T]} \cup \{\mathbf{x}\}$
8:          $\mathbf{Y}_i^{[\#T]} \leftarrow \mathbf{Y}_i^{[\#T]} \cup \{y(G)\}$
9: **function** INFER$(G)$
10:     $\mathbf{z} \leftarrow \left\{ \mathcal{F}_i^{[\#T]}(G, \mathbf{x}) \;\middle|\; H_i \in \mathcal{H} \text{ and } (T, \mathbf{x}) \leftarrow H_i(G) \right\}$
11:     **return** $\widehat{y}(\mathbf{z})$

---

### 2.5 ONLINE LEARNING OF TEMPLATE HIERARCHIES

**Outline.** Given a stream of graphs $\mathcal{D} = (G_1, G_2, \dots)$, we design an algorithm that can make prediction on every $G_j \in \mathcal{D}$ using all prior $\{G_k \in \mathcal{D} \mid k < j\} \triangleq \mathcal{D}_{<j}$. The main idea is to learn many (simple) models. **All graphs whose templates are isomorphic share the same model.** Each $H_i \in \mathcal{H}$ processes every $G \in \mathcal{D}$. Templatizer $H_i(G)$ extracts graph template $T$ and features $\mathbf{x}$. For **inference**, model associated with $\#_T$ is retrieved[2], then invoked to predict $y$ from $\mathbf{x}$. All $|\mathcal{H}|$ predictions can be combined with high-level master model $\widehat{y} : \mathbb{R}^{|\mathcal{H}|} \to \mathcal{Y}$. For **training**, once the ground-truth answer $y(G)$ is retrieved, models update to learn from $(\mathbf{x}, y(G))$.

**Feature-Label Matrices per (templatizer, template)-Pair.** Given templatizer $H_i \in \mathcal{H}$, an arbitrary template $T$ produced by $H_i$, and timestamp $j \leq |\mathcal{D}|$, then the set

$$\mathbf{X}_{i,<j}^{[\#T]} = \{ \mathbf{x}_k \mid (T_k, \mathbf{x}_k) \leftarrow H_i(G_k) \text{ if } \#_{T_k} = \#_T \}_{G_k \in \mathcal{D}_j} \tag{7}$$

Can be cast as matrix, since its rows of $\mathbf{x}_j \in \mathbb{R}^{d_T}$ are of the same[3] dimensionality, whose graph templates are isomorphic. Hence, $\mathbf{X}_{i,<j}^{[\#T]} \in \mathbb{R}^{\circ \times d_T}$ and label matrix

$$\mathbf{Y}_{i,<j}^{[\#T]} = \{ y(G_k) \mid \#_T = \#_{H_i(G_k)} \}_{G_k \in \mathcal{D}_{<j}} \in \mathcal{Y}^{\circ} \quad \text{where} \quad \circ = \sum_{k<j} \mathbf{1}_{[\#_T = \#_{H_i(G_k)}]}. \tag{8}$$

**Model per (templatizer, template)-Pair.** Let $\mathcal{F}_i^{[\#_T]} : \mathbb{R}^{d_T} \to \mathcal{Y}$ denote model specialized for template $T$ of $H_i$. There are many possibilities for $\mathcal{F}_i^{[\#_T]}$, which we co-design with corresponding $H_i$ (see §3.3). **Inference** on subsequent $G$ can run $|\mathcal{H}|$ (parallel) invocations:

$$\mathbf{z}_G = \left\{ \mathcal{F}_i^{[\#_T]}(\mathbf{x}) \mid H_i \in \mathcal{H} \text{ and } (T, \mathbf{x}) \leftarrow H_i(G) \right\} \in \mathbb{R}^m. \tag{9}$$

then invoke $\widehat{y}(\mathbf{z}_G)$. For **training**, some models $\mathcal{F}_i^{[\#_T]}$ update periodically using $\left( \mathbf{X}_{i,<j}^{[\#T]}, \mathbf{Y}_{i,<j}^{[\#T]} \right)$, while others incrementally absorb each incoming observation $(G_j, y(G_j))$ – see, §3.3. Nonetheless, learning can happen in parallel for all $\mathcal{F}_i^{[\#_T]}$.

**Algorithm 1** defines routines (ADDEXAMPLE, INFER), initializes master model $\widehat{y}$, and initializes data structure $\mathcal{F}$ to a Hashtable. At every $G_j \in \mathcal{D}$, routine INFER$(G_j)$ can return the estimated quantity of interest (*e.g.*, cardinality), by invoking $\widehat{y}$ on the output of $|\mathcal{H}|$ invocations of $\mathcal{F}$. Once the caller retrieves the ground-truth value $y(G_j)$ (*e.g.*, as the query results are assembled) then routine ADDEXAMPLE can incorporate the example $(G_j, y(G_j))$ into the (simple) models within $\mathcal{F}$.

---

[2]All models are small (kept in RAM). In practice, since probability of false collision is low albeit non-zero, the *actual* hashtable keys we use are $(\#_T, d_T)$, i.e., pairing with dimension of the $\mathbf{x}$ produced alongside $T$.

[3]$d_T = \sum_{v \in \mathcal{V}} \sum_{(j, \psi) \in \mathbf{S}_x^{(v)}} d_\psi$.

# 3 HIERARCHICAL GRAPH LEARNERS FOR CARDINALITY ESTIMATION

## 3.1 TEMPLATIZATION

We studied three templatization strategies, $H_1, H_2, H_3$, ranging from fine-grained to coarse-grained templates. Table 1 shows the feature sets kept in the template $T$ VS extracted to the dense vector $\mathbf{x}$, for every $H_i$. For example, The fine-grained $H_1$ removes just the {constant value} from the template. Hence query graphs found in the same $H_1$ template differ only by the constant values.

Table 1: Templatization Strategies. Each $H_i$ templatizes as $(T_i, \mathbf{x}_i) \leftarrow H_i(G)$ where $T$ and $\mathbf{x}$ include features listed in, respectively, $\mathbf{S}_t$ and $\mathbf{S}_x$. The choices $\mathbf{S}_t$ **induce a divisive hierarchy** as every $\mathbf{S}_t$ row includes the information of the next row (column name determines column type).

| Templatizer | Hash features $\mathbf{S}_t$ | Dense (model) features $\mathbf{S}_x$ |
|---|---|---|
| $H_1$ | {Table name, column name, predicate op} | {constant value} |
| $H_2$ | {Table name, column name} | {constant value, predicate op} |
| $H_3$ | {Table name, column type} | {constant value, predicate op, column unique value} |

## 3.2 FEATURIZERS

The templatizer extracts features of many types into $\mathbf{x}$, including numeric, string, date, time, boolean, respectively, we use featurizers $\psi$ as, identity, ASCII of first-3 characters (in base 256), as numeric YYYYMMDD, as numeric hhmmss, as $\{0, 1\}$. Finally, we map each into the range $[0, 1]$. We explore two scaling techniques: normalizing (ie. $\frac{v - \min}{\max - \min}$) and replacing with percentile. We map predicate operators $(>, <, =, \text{or}, \text{and}, \dots)$ to unique integers.

Further, we add one more feature that our models find useful: combining combine the constant with the predicate operator to produce range vector. For example, "$\leq 2000$" is featurized as $[0, 0.3]$, "$= 2000$" becomes $[0.3, 0.3]$, and "$\geq 2000$" becomes $[0.3, 1]$ (supposing constant 2000 scales to 0.3); all other predicate operators are currently featurized as $[0, 1]$ for simplicity.

## 3.3 LEARNING

We use a rule-based $\widehat{y}$. Its output $\widehat{y}(G)$ can be concisely described with a flow-chart:

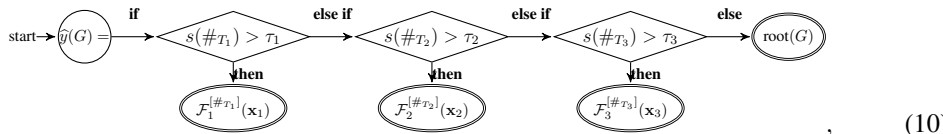

$$\tag{10}$$

where the history size $s(\#_{T_i})$ equals the number of observations that hash to $\#_{T_i}$, *i.e.*, the height of matrices $\mathbf{X}_i^{[\#_{T_i}]}$ and $\mathbf{Y}_i^{[\#_{T_i}]}$. Given graph $G \in \mathcal{G}$, let $(T_i, \mathbf{x}_i) \leftarrow H_i(G)$ for $i \in \{1, 2, 3\}$. Further, let $\tau_1 < \tau_2 < \tau_3$ denote "*activation thresholds*"[4]. If the size of $\#_{T_i}$ meets the threshold $\tau_i$, we invoke the corresponding $\mathcal{F}_i$. If not, we move on to the next hierarchy level.

- root$(G)$ will be invoked when incoming query $G$ has an unfamiliar template (*a.k.a*, the cold-start problem). We propose to set root$(G)$ to a default estimator, eg. Postgres.
- We try-out several choices for $\mathcal{F}$. All showing incremental and/or instant training, *e.g.*,

$$\textbf{Linear Regression:} \quad \mathcal{F}_i^{[\#T]}(\mathbf{x}) = \left(\mathbf{X}_i^{[\#T]}\right)^\dagger \mathbf{Y}_i^{[\#T]} \mathbf{x} \tag{11}$$

$$\textbf{Gaussian Kernel:} \quad \mathcal{F}_i^{[\#T]}(\mathbf{x}) = \frac{\sum_j \mathbf{Y}_{i,j}^{[\#T]} e^{-d(\mathbf{x}, \mathbf{X}_{i,j}^{[\#T]})}}{\sum_j e^{-d(\mathbf{x}, \mathbf{X}_{i,j}^{[\#T]})}} \tag{12}$$

$$\textbf{Gradient-Boosted Decision Trees:} \text{ implementation of (Chen \& Guestrin, 2016)} \tag{13}$$

---

[4] It will always be that $s(\#_{T_1}) < s(\#_{T_2})$, due to the divisive hierarchy

where $(.)^\dagger$ denotes Moore-Penrose inverse and $d(.,.)$ denotes distance function (see **Appendix**). For linear regression (Eq. 11), we add[5] column of 1 to $\mathbf{x}$ and to $\mathbf{X}_i^{[\#T]}$. As observations $\left(\mathbf{X}_i^{[\#T]}, \mathbf{Y}_i^{[\#T]}\right)$ grow, it is unnecessary to re-compute (from scratch) the pseudo-inverse $(.)^\dagger$. It can be incrementally updated, *e.g.*, with rank-1 changes to the Singular Value Decomposition of $\mathbf{X}_i^{[\#T]}$, per Brand (2006).

## 4 EXPERIMENTAL EVALUATION

**Metrics.** We quantify the error of cardinality estimate $\widehat{y}(G)$ and true (label) cardinality $y(G)$ with:

$$Q_{\text{err}} = \max\left(\frac{y}{\widehat{y}}, \frac{\widehat{y}}{y}\right) \quad (14) \qquad A_{\text{err}} = |\widehat{y} - y| \quad (15) \qquad R_{\text{err}} = 1 - \frac{\min(\widehat{y}, y)}{\max(\widehat{y}, y)} \quad (16)$$

respectively known as Q-error, absolute error, and relative error.

**Datasets.** We run experiments on several database workloads, downloaded from benchmark (Cardbench, Chronis et al., 2024) (prefixed "binaryjoin-" within figures). Further, we extend their query generator to: 1) enable multi-way join queries (up-to 5 joins) to increase the query complexity; 2) incorporate the high repetiveness feature of data warehouse workloads as in Redshift (van Renen et al., 2024) (prefixed "multijoin-"). For all multijoin datasets, we fixed the sample constant size at 10 and varied the sample size (repetition rate) to evaluate its impact on accuracy in Fig 4.

**Models.** We use Cardinality Estimation models – Postgres, MSCN, ours $\{H_i, \mathcal{F}_i\}_i$.

(1) **Postgres**: Traditional histogram-based estimator implemented in open-source PostgreSQL (PostgreSQL Group). This estimator can be invoked on any query (100% admit rate).

(2) **MSCN**: Neural-based estimator (Kipf et al., 2019). We train two model copies, per database workload: "MSCN" and "MSCN+", respectively, on 1000 query graphs and on 25% of the graphs (3.3X-10X vs MSCN). Crucially, MSCN cannot admit queries containing "or" predicates[6]. On our workloads, MSCN admits 61% of the queries.

(3) **Ours**: History-based estimator. We infer using $(\mathcal{F}_i, H_i)$ per Eq.11–13, either for singular $i = 1$ or multiple $\{(\mathcal{F}_i, H_i)\}_{i \in \{1,2,3\}}$ that live on a hierarchy (§2.5). Singular $(\mathcal{F}_i, H_i)$ can estimate only if there are enough observations of template $\{T_i \leftarrow H_i(G)\}$.

**Overview.** We conduct three kinds of experiments: §4.1 evaluates the **practical scenario** that **inference is required** for all queries. Here, a method can fall-back onto another. §4.2 conducts apples-to-apples comparison of our models against prior work; §4.3 Ablates our models;

### 4.1 HIERARCHICAL MODELS

In this set of experiments, methods **must always make a prediction**. Our method defaults to the Postgres estimator, in cases, where the graph structure is novel (has not appeared earlier in the online setting). Our full hierarchy, depicted in Figure 1b and formalized in Equation 10, is abbreviated $(H_1, H_2, H_3, \mathbb{P})$, where $\mathbb{P}$ denoting Postgres estimator. We set thresholds $(\tau_1, \tau_2, \tau_3)$ in Eq.10 to (3, 10, 100) and employ Gradient-Boosted Decision Trees (GBDT) at each hierarchical level.

**How effective are hierarchical learners?** Table 2 compares hierarchical models with different hierarchy combinations. Comparing $(H_1, H_2, H_3, \mathbb{P})$, $(H_2, H_3, \mathbb{P})$, $(H_3, \mathbb{P})$, and Postgres, we can see the models keep improving when we add more levels of hierarchy and the full hierarchy of models is always better than Postgres at all metrics. In addition, The full hierarchy leverages each level effectively, as evidenced by the activation ratios (0.69, 0.04, 0.01, 0.26) for $H_1$, $H_2$, $H_3$, and Postgres, respectively. These results demonstrate the effectiveness of our hierarchical models in leveraging historical data to enhance the cardinality estimation capabilities of traditional optimizers.

**The necessity of multiple hierarchy?** Table 2 also shows the need of hierarchy. Comparing $(H_1, \mathbb{P})$, $(H_1, H_2, \mathbb{P})$, $(H_1, H_2, H_3, \mathbb{P})$, the latter two consistently outperform the first. This indicates that a simple hierarchy $(H_1, \mathbb{P})$ is insufficient, highlighting the importance of multi-level hierarchies.

---

[5]Equivalent to adding bias-term to one-layer model.

[6]As-is, MSCN (Kipf et al., 2019) was developed for conjunctions only, its extension is beyond our scope.

Table 2: Hierarchical models. Median relative error, median absolute error and Q-Error percentiles.

| hierarchy | $R_\text{err}$ | $A_\text{err}$ | $Q_\text{err}^{50}$ | $Q_\text{err}^{90}$ | $Q_\text{err}^{95}$ | $R_\text{err}$ | $A_\text{err}$ | $Q_\text{err}^{50}$ | $Q_\text{err}^{90}$ | $Q_\text{err}^{95}$ |
|---|---|---|---|---|---|---|---|---|---|---|
| **multijoin-cms** | | | | | | **multijoin-stackoverflow** | | | | |
| Postgres | 0.70 | $2.4e^5$ | 3.33 | 112 | $2.3e^3$ | 0.79 | $2.8e^5$ | 4.85 | 360 | $3.1e^3$ |
| $(H_3, \text{P})$ | 0.69 | $2.2e^5$ | 3.21 | 110 | $2.2e^3$ | 0.77 | $1.8e^5$ | 4.30 | 367 | $3.8e^3$ |
| $(H_2, H_3, \text{P})$ | 0.13 | $2.0e^4$ | 1.15 | 46.67 | 159 | 0.14 | $1.7e^3$ | 1.16 | 44.33 | 464 |
| $(H_1, \text{P})$ | 0.06 | $9.1e^3$ | 1.07 | 22.22 | 97.00 | 0.10 | 456 | 1.12 | 21.03 | 200 |
| $(H_1, H_2, \text{P})$ | **0.06** | $\mathbf{8.5e^3}$ | **1.06** | **20.10** | **94.48** | **0.10** | **388** | **1.11** | **18.01** | **182** |
| $(H_1, H_2, H_3, \text{P})$ | **0.06** | $\mathbf{8.5e^3}$ | **1.06** | **20.10** | **94.48** | **0.10** | **388** | **1.11** | **18.01** | **182** |
| **multijoin-accidents** | | | | | | **multijoin-airline** | | | | |
| Postgres | 0.39 | $8.8e^7$ | 1.65 | 10.31 | 18.29 | 0.39 | $2.6e^4$ | 1.63 | 97.30 | 216 |
| $(H_3, \text{P})$ | 0.25 | $3.1e^7$ | 1.34 | 8.93 | 20.60 | 0.37 | $2.4e^4$ | 1.59 | 97.00 | 216 |
| $(H_2, H_3, \text{P})$ | 0.13 | $1.2e^7$ | 1.15 | **4.81** | **15.42** | 0.17 | $6.0e^3$ | 1.20 | 13.88 | 91.00 |
| $(H_1, \text{P})$ | 0.13 | $1.1e^7$ | 1.15 | 4.95 | 17.25 | 0.12 | $3.2e^3$ | 1.13 | 4.50 | 29.20 |
| $(H_1, H_2, \text{P})$ | **0.13** | $\mathbf{1.1e^7}$ | **1.15** | 5.02 | 17.70 | **0.12** | $\mathbf{3.1e^3}$ | **1.13** | **4.29** | **25.00** |
| $(H_1, H_2, H_3, \text{P})$ | **0.13** | $\mathbf{1.1e^7}$ | **1.15** | 5.02 | 17.70 | **0.12** | $\mathbf{3.1e^3}$ | **1.13** | **4.29** | **25.00** |
| **multijoin-employee** | | | | | | **multijoin-geo** | | | | |
| Postgres | 0.35 | $1.2e^3$ | 1.54 | 3.38 | 4.83 | 1.00 | $9.2e^6$ | 224 | $2.1e^5$ | $1.2e^6$ |
| $(H_3, \text{P})$ | 0.26 | 961 | 1.35 | 3.14 | 4.42 | 1.00 | $8.9e^6$ | 218 | $2.1e^5$ | $1.2e^6$ |
| $(H_2, H_3, \text{P})$ | 0.04 | 481 | 1.05 | 2.11 | **2.98** | 0.09 | $1.6e^4$ | 1.10 | $5.8e^3$ | $7.3e^4$ |
| $(H_1, \text{P})$ | 0.03 | 297 | 1.03 | 2.09 | 3.07 | 0.08 | $4.3e^3$ | 1.09 | 192 | $1.1e^4$ |
| $(H_1, H_2, \text{P})$ | **0.03** | **269** | **1.03** | **2.03** | 3.01 | **0.07** | $\mathbf{3.3e^3}$ | **1.08** | **66.38** | $\mathbf{7.0e^3}$ |
| $(H_1, H_2, H_3, \text{P})$ | **0.03** | **269** | **1.03** | **2.03** | 3.01 | **0.07** | $\mathbf{3.3e^3}$ | **1.08** | **66.38** | $\mathbf{7.0e^3}$ |
| **binaryjoin-stackoverflow** | | | | | | **binaryjoin-airline** | | | | |
| Postgres | 0.69 | $1.5e^7$ | 3.28 | 160 | 470 | 0.55 | $9.3e^4$ | 2.22 | 37.17 | 127 |
| $(H_3, \text{P})$ | 0.66 | $9.0e^6$ | 2.93 | 149 | 382 | 0.53 | $2.0e^5$ | 2.11 | 63.00 | 206 |
| $(H_2, H_3, \text{P})$ | 0.42 | $1.8e^6$ | 1.74 | 60.48 | 183 | 0.45 | $1.5e^5$ | 1.82 | 51.55 | 190 |
| $(H_1, \text{P})$ | 0.43 | $1.7e^6$ | 1.76 | 53.33 | 175 | 0.44 | $5.3e^4$ | 1.80 | **28.15** | **112** |
| $(H_1, H_2, \text{P})$ | 0.40 | $1.5e^6$ | 1.66 | **44.00** | **174** | **0.44** | $\mathbf{5.3e^4}$ | **1.80** | 28.24 | 112 |
| $(H_1, H_2, H_3, \text{P})$ | **0.39** | $\mathbf{1.5e^6}$ | **1.63** | 45.15 | 175 | 0.44 | $1.4e^5$ | 1.80 | 43.00 | 179 |

## 4.2 COMPARING INDIVIDUAL MODELS

In this section, we compare all methods **on the intersection** of queries they are able to admit – about 25% of queries. While §4.1 shows practical hierarchies that are able to process any query, this provides a sound apples-to-apples comparison.

Table 3 summarizes the performance of four models: Postgres, MSCN, MSCN+, and only one model-templatizer pair $(\mathcal{F}_1, H_1)$, specifically, GBDT (Eq. 13) with $H_1$. MSCN+ (trained on $\approx$ 5X more data) is much better than MSCN and is frequently better than Postgres. Overall, our method is competitive and produces higher accuracy majority of the time. In particular, $H_1$ is substantially better (10X-50X+) than Postgres half-of-the-time. We also observe that our model is more robust at the tail of the error distribution (P90 and P95).

## 4.3 ABLATION STUDIES

**Model Choice.** We compare across choices of models $\mathcal{F}$ (Eq. 11–13) and $H_i$'s in Fig 3. We find that Gradient-Boosted Decision Trees (GBDT) are consistently strong across different datasets and level of hierarchy, so we choose GBDT for Table 3, and on every level of hierarchy in Table 2.

**Repetition Rate.** We modify the workload generator in Chronis et al. (2024) to enable more constants for each predicate in the query. For example, instead of generating a query with predicates "a > 5 AND b = 2", our modified generator will generate "a > 5 AND b = 2", "a > 5 AND b = 20", "a > 1 AND b = 2", "a > 1 AND b = 20" when the sample size is 2, meaning that each predicate will have 2 constants to choose from (ie. a > [1, 10], b = [2, 20]). The constant sample sizes in the experiment we choose are [1, 3, 10], therefore it generates the repetition rate of 20%, 81% and 91% in query templates. As shown in Fig 4, all templatization strategies exhibit improved performance with increasing workload repetition, while maintaining low q-error levels.

Table 3: Model Errors at various percentiles, per dataset. We **bold** strongest number per (database, q-error percentile).

| model | $R_{\text{err}}$ | $A_{\text{err}}$ | $Q_{\text{err}}^{50}$ | $Q_{\text{err}}^{90}$ | $Q_{\text{err}}^{95}$ | $R_{\text{err}}$ | $A_{\text{err}}$ | $Q_{\text{err}}^{50}$ | $Q_{\text{err}}^{90}$ | $Q_{\text{err}}^{95}$ |
|---|---|---|---|---|---|---|---|---|---|---|
| | **multijoin-cms** | | | | | **multijoin-stackoverflow** | | | | |
| postgres | 0.68 | $2.5e^5$ | 3.17 | 53.79 | $3.0e^3$ | 0.74 | $4.2e^4$ | 3.87 | 149 | $1.6e^3$ |
| MSCN | 0.56 | $3.7e^5$ | 2.28 | 13.06 | 30.33 | 0.89 | $1.6e^4$ | 8.86 | 62.77 | 167 |
| MSCN+ | 0.41 | $2.4e^5$ | 1.69 | 4.65 | 7.08 | 0.50 | $9.7e^3$ | 2.00 | 10.99 | 30.98 |
| $H_1$ | **0.02** | $\mathbf{5.1e^3}$ | **1.02** | **1.69** | **2.74** | **0.05** | **30.03** | **1.05** | **2.19** | **5.46** |
| | **multijoin-accidents** | | | | | **multijoin-airline** | | | | |
| postgres | 0.42 | $6.0e^7$ | 1.73 | 11.04 | 20.21 | 0.20 | $1.3e^5$ | 1.25 | 8.68 | 44.04 |
| MSCN | 0.74 | $5.4e^7$ | 3.82 | 17.82 | 46.64 | 0.37 | $3.0e^5$ | 1.59 | 7.71 | 14.09 |
| MSCN+ | 0.54 | $3.1e^7$ | 2.20 | 8.31 | **15.22** | 0.39 | $3.3e^5$ | 1.65 | 7.28 | 12.32 |
| $H_1$ | **0.08** | $\mathbf{4.0e^6}$ | **1.09** | **3.31** | 19.13 | **0.11** | $\mathbf{6.6e^4}$ | **1.13** | **3.24** | **8.98** |
| | **multijoin-employee** | | | | | **multijoin-geo** | | | | |
| postgres | 0.35 | $2.6e^3$ | 1.53 | 3.46 | 5.26 | 0.99 | $4.7e^6$ | 161 | $1.7e^5$ | $9.0e^5$ |
| MSCN | 0.38 | $1.9e^4$ | 1.61 | 4.20 | 7.18 | 0.51 | $8.0e^3$ | 2.03 | 9.87 | 15.50 |
| MSCN+ | 0.17 | $7.1e^3$ | 1.20 | 1.74 | **2.10** | 0.49 | $7.1e^3$ | 1.98 | 5.68 | 9.35 |
| $H_1$ | **0.01** | **268** | **1.01** | **1.59** | 2.23 | **0.02** | **99.00** | **1.02** | **1.67** | **3.05** |
| | **binaryjoin-stackoverflow** | | | | | **binaryjoin-airline** | | | | |
| postgres | 0.68 | $2.4e^7$ | 3.16 | 161 | 332 | 0.38 | $2.2e^6$ | 1.62 | 6.43 | 23.40 |
| MSCN | 0.41 | $2.6e^6$ | 1.68 | 10.96 | 29.36 | 0.73 | $3.1e^6$ | 3.71 | 60.08 | 91.24 |
| MSCN+ | 0.29 | $2.3e^6$ | 1.41 | 3.34 | 5.43 | 0.79 | $3.5e^6$ | 4.78 | 39.58 | 45.57 |
| $H_1$ | **0.02** | $\mathbf{1.2e^5}$ | **1.02** | **1.41** | **2.14** | **0.01** | $\mathbf{2.2e^4}$ | **1.01** | **1.32** | **1.70** |

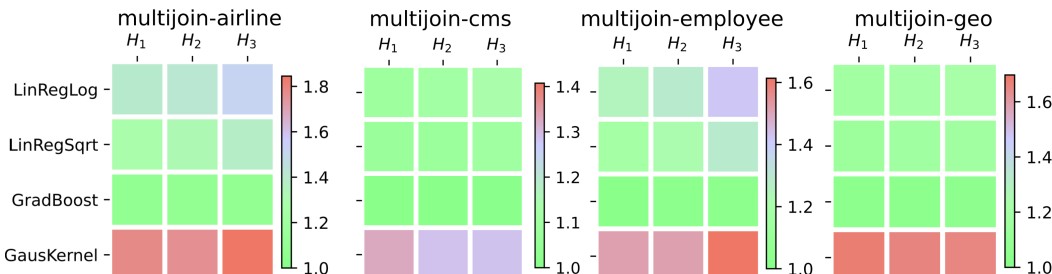

Figure 3: 50th percentile Q-error per database, comparing templatization strategies and learners.

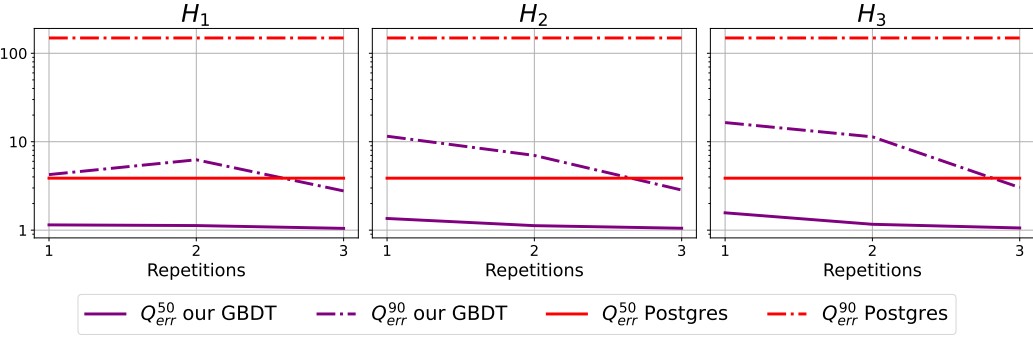

Figure 4: Accuracy of our learners, as a function of repetition amount. Each chart shows one templatization strategy, containing 4 lines: {Gradient Boosted Decision Tree (Eq. 13), Postgres Estimator } $\times$ {50th, 90th Q-errors}. The Y-axis displays Q-errors.

**History Size.** We assess the performance of learners as a function of *history size*, in the Appendix.

## 5 RELATED WORK

**Learned Cardinality Estimation.** In the recent years, several lines of approach learned cardinally estimation have been proposed (Han et al., 2021; Sun et al., 2021; Kim et al., 2022). The first line is workload-driven learning (Kipf et al., 2019; Negi et al., 2023; Reiner & Grossniklaus, 2024), which requires pre-collected workload queries and their executions against the database to collect true cardinalities as the training data. To reduce cost of acquiring training data, the second direction explores data-driven learning (Yang et al., 2019; 2021; Hilprecht et al., 2020; Wu et al., 2023; Kim et al., 2024), which learns a model only on the data capturing its distributions without running any queries. While these models do not have the overhead of running queries, for large databases it could still take hours to train such models. Kim et al. (2024) develops auto-regressive model that samples queries matching filters, crucially supporting string and disjunctive filters. Another line includes localized-models which learn lightweight models that can to capture certain query patterns and can adapt online. Our own work falls into this category. Our method is most-similar to (Malik et al., 2007), since they also group queries by templates, and also do learning-and-inference on dense-vectors within each template. However, we differ in two ways: (1) The templates of (Malik et al., 2007) use a flat vector representation for queries, our are graphs and for grouping we use graph hashes – as such, ours are invariant to node orderings (2) We learn hierarchies of models rather than a flat grouping of models. Moreover, other approaches have explored also different directions to represent queries for localized models. For comparison, Dutt et al. (2019) creates conjunction trees made of simple predicates while Woltmann et al. (2019) learn models on groups of related tables. All these representations are less expressive than query graphs to provide a direct way to represent queries in databases. In fact, our modeling approach to represent queries is very similar to methods used to learn query cost prediction (*e.g.*, execution time) (Hilprecht & Binnig, 2022; Wu et al., 2024) which also uses a query graph representation while our approach uses them to represent groups of similar queries for cardinality estimation.

**Graph Hashing.** Helbling (2020) compute hash values for directed graphs, also by extending Merkle Trees (Merkle, 1988). There are also other methods that can operate on directed but also undirected graphs, including (Portegys, 2008) and WL (Shervashidze et al., 2011). These methods iteratively update node's hash using itself and its neighbors. Each update-round incorporates information from further neighbors. The number of iterations could be set to the graph diameter. Our algorithm slightly differs as our graph nodes *could* be invariant neighbor orders *sometimes* (*e.g.*, *or* junction), while being variant at *other times* (*e.g.*, $>$ operator). In addition, we only work with DAGs and therefore iterating in topological order terminates the algorithm.

**Decoupled Graph Neural Nets.** Our method is also linked to methods that "*decouple*" the graph-processing step from the learning. Specifically, methods that extract features using the graph and no longer need the graph for learning. These methods include (Wu et al., 2019; Frasca et al., 2020). In that regard, our method also uses the graph for pre-processing. We differ than those methods as they use the structure to propagate information along edges whereas we hash the structure.

## 6 CONCLUSION

In this paper, we propose a localized on-line models for cardinality estimation. Queries with isomorphic structures will be grouped-together, with different templatization strategies forming a hierarchy. Within each group, a simple model, *e.g.*, linear regression or gradient-boosted decision trees, can be trained to estimate cardinality of a given query. A predictions is always made at the lowest-level node with sufficient observations, and falls back onto either neural or traditional methods at the root. However, this new query already establishes an observation when the pattern is repeated. In the experiments, we show that our models outperform traditional and neural models, and produce robust accuracy even at the tail (P90 and P95). Moreover, $H_1$ is substantially better (10X-50X+) than Postgres half-of-the-time. As future work, we plan to explore different grouping methods, increasing the hierarchy with more templatization strategies, and explore different default models.

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

# A APPENDIX

## A.1 DATASET STATS

This section presents the statistics of the datasets used in this paper. Importantly, Table 4 presents the repetition rates at different template levels, following the definition from van Renen et al. (2024). Our multi-join workloads, with $H_1$ repetition rates between 83% and 96%, closely mimic the 90% template repetition rate reported in van Renen et al. (2024). Table 5 summarizes the diverse databases used in our experiments. The smallest databases (accidents and employee) have 3 and 6 tables, respectively, while the largest database (cms_synthetic_patient_data_omop) comprises 24 tables and 32 billion rows.

Table 4: Workload Statistics.

| Workload | Database | # Queries | Repetition Rate (%) | | |
| --- | --- | --- | --- | --- | --- |
| | | | $H_1$ | $H_2$ | $H_3$ |
| multijoin-stackoverflow | stackoverflow | 16k | 91 | 94 | 95 |
| binaryjoin-stackoverflow | | 13k | 67 | 85 | 96 |
| multijoin-airline | airline | 20k | 93 | 95 | 96 |
| binaryjoin-airline | | 13k | 34 | 56 | 94 |
| multijoin-accidents | accidents | 29k | 95 | 97 | 98 |
| multijoin-cms | cms_synthetic_patient_data_omop | 14k | 83 | 87 | 88 |
| multijoin-geo | geo_openstreetmap | 13k | 94 | 96 | 96 |
| multijoin-employee | employee | 62k | 96 | 98 | 98 |

Table 5: Database Statistics.

| Database | # Tables | # Columns | # Rows | # Join Paths |
| --- | --- | --- | --- | --- |
| stackoverflow | 14 | 187 | 3.0B | 13 |
| airline | 19 | 119 | 944.2M | 27 |
| accidents | 3 | 43 | 27.4M | 2 |
| cms_synthetic_patient_data_omop | 24 | 251 | 32.6B | 22 |
| geo_openstreetmap | 16 | 81 | 8.3B | 15 |
| employee | 6 | 24 | 48.8M | 5 |

## A.2 HASHING FUNCTION EXTENDED

In this section, we includes the algorithm (Algorithm 2) and comparison table (Table 6) to further illustrate the hashing function in Section 2.4.

Table 6: Input data requirements. Merkle's method is designed for balanced search trees (BSTs), with features only on leaf nodes. Our generalization (Alg. 2) produces identical output to Merkle's when input is BST, additionally generalizing to DAG inputs.

| Comparison | Merkle Trees (Merkle, 1988) | DAG Hashing (Alg. 2) |
| --- | --- | --- |
| Hashable Structure is: | Tree (w/ virtual edges) | DAG (edges from query graph) |
| Input Data (features) are on: | only leaf nodes | all nodes |
| Neighbors are: | always ordered | can be order-invariant |

## A.3 ABLATION STUDIES EXTENDED

We also conduct ablation experiments to show that, in general, our simple models improve as data accumulates in each template (Fig. 5). As $H_1$ is the most-grained, it stabilizes earlier and has

---

**Algorithm 2** Hashing function $\# : \mathcal{G} \to \{0,1\}^h$ for Directed Acyclic Graphs (DAGs).

---

1: **input:** hashing function of bit-vectors ($\$ : \{0,1\}^* \to \{0,1\}^h$), *e.g.*, MD5 (Rivest, 1992).
2: **input:** Directed Acyclic Graph $T = (\mathcal{V}, \mathcal{E}, f)$.
3:
4: **for** $v \in \mathcal{V}$ **do**
5:     $\mu_v \leftarrow \$(f^{(v)})$
6: **for** $v \in \pi$ **do**   // process in topological order
7:     **if** operation $v$ is invariant to order of predecessors **then**
8:         $\mu_v \leftarrow \$(\mu_v || \textsc{UnorderedCombine}(\{\mu_u \mid (u,v) \in \mathcal{E}\})$
9:     **else** // Sometimes, order matters. E.g., A > B differs from B > A
10:         $\mu_v \leftarrow \$(\mu_v || \textsc{OrderedCombine}(\{\mu_u \mid (u,v) \in \mathcal{E}\})$
11: $\pi^* \leftarrow \textsc{DeterministicTopologicalOrder}(T, \mu)$
12: **return** $\$\,(\textsc{OrderedCombine}(\{\mu_v \mid v \in \pi^*\})$
13:
14: **function** $\textsc{OrderedCombine}(\{z \in \{0,1\}^h\})$
15:     **return** $\textsc{Concat}(z)$
16: **function** $\textsc{UnorderedCombine}(\{z \in \{0,1\}^h\})$
17:     **return** $\textsc{Concat}(\texttt{sorted}(z))$
18: **function** $\textsc{DeterministicTopologicalOrder}(G, \mu)$
19:     $\pi^* \leftarrow [\,]$
20:     $\textsc{UnprocessedPrev}_v \leftarrow \{u \mid (u,v) \in \mathcal{E}\}$, for all $v \in \mathcal{V}$
21:     **while** $\pi^*.\texttt{size} < \mathcal{V}.\texttt{size}$ **do**
22:         $q \leftarrow q \cup \{(\mu_v, v) \mid v \in \mathcal{V} \textbf{ if } \textsc{UnprocessedPrev}_v = \emptyset \textbf{ and } v \notin \pi^*\}$
23:         $(\_, u) \leftarrow \max(q)$
24:         $\pi^*.\textsc{Append}(u)$
25:         **for** $v \in \{v' \mid (u, v') \in \mathcal{E}\}$ **do**
26:             $\textsc{UnprocessedPrev}_v \leftarrow \textsc{UnprocessedPrev}_v \backslash \{u\}$

---

lower tail errors. Notably, the accuracy of coarser templatization, *e.g.*, $H_3$, combining records from multiple (columns, predicate operators), needs more training history data to converge. It also shows that GBDT always has better performance than Linear Regression (LR) and Gaussian Kernel(GK) models accross different datasets. This also matches our observation in Figure 3.

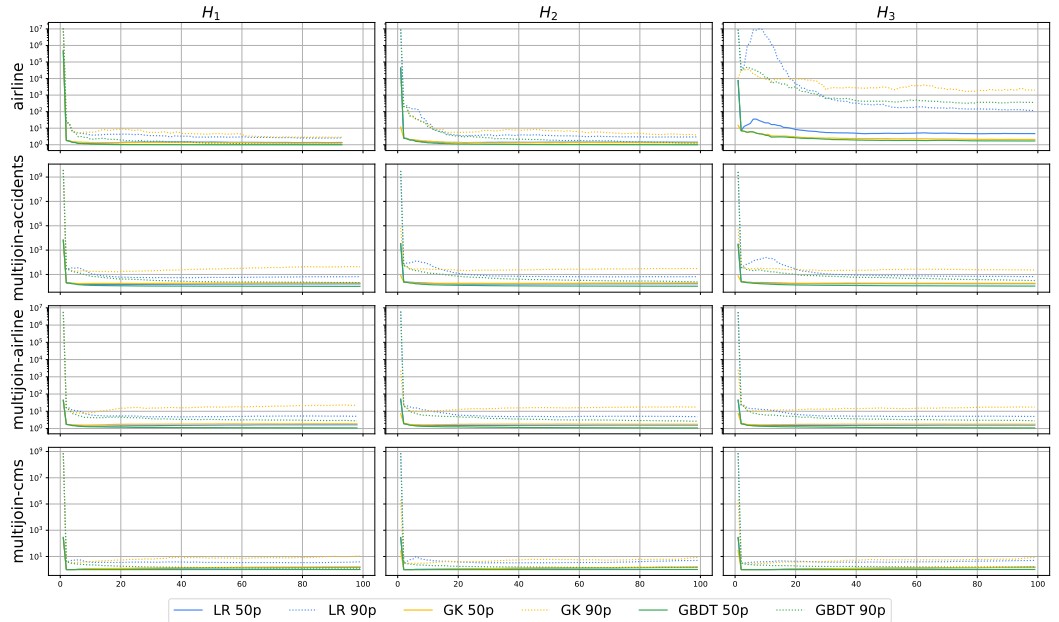

Figure 5: Each subplot shows Q-error percentiles as function of amount of history per workload & templatization strategy. In particular, each line color represents learner (Eq.11–13) and each line style represents percentile. History size is less than or equal to x-axis value.

