# OpenReview forum: "Hierarchical Graph Learners for Cardinality Estimation"
_ICLR.cc/2025/Conference — Submitted to ICLR 2025_

### Official Review · Reviewer_2Nam · 2024-11-01

**Soundness:** 3
**Presentation:** 2
**Contribution:** 2
**Rating:** 5
**Confidence:** 4

**Summary:**

To address the issue of repeated queries in cardinality estimation, this paper proposes an on-line cardinality estimation method. Unlike traditional cardinality estimation approaches that rely on a single model, this study introduces a hierarchical cardinality estimation framework. Queries are categorized into three levels, with different structural classifications applied at each level. For each level, distinct estimator models are trained based on various classification templates, and evaluations are conducted hierarchically. Additionally, these models utilize an extension of Merkle-Trees to hash directed acyclic graph (DAG) query plans. Finally, an ensemble learning method is used to statistically aggregate the results and produce the final cardinality estimates. Compared to traditional cardinality estimators and the query-based cost estimation method MSCN, this approach achieves superior results.

**Strengths:**

S1. Hierarchical cardinality estimation methods have significant advantages for cardinality estimation methods in the presence of a large number of repeated queries. They allow for the training of separate cardinality estimators for each query type, thereby saving training costs and improving the accuracy of cardinality estimation.
S2. The method presented in this paper demonstrates strong cardinality estimation performance, and it also exhibits good convergence stability through hierarchical training.
S3. Compared to traditional cardinality estimators, the method proposed in this paper achieves faster convergence speed with lower overhead, making it suitable for practical industrial applications.

**Weaknesses:**

W1. This paper employs only the Q-Error as an evaluation metric, which can assess the stability of the cardinality estimator but does not provide an intuitive measure of its accuracy. The addition of mean absolute error (MAE) and relative prediction error (RPE) would allow for a more comprehensive evaluation of the accuracy of different cardinality estimators.
W2. This paper compares only with two relatively outdated query-based cardinality estimation methods, MSCN and MSCN+. It should include a broader variety and greater number of baseline methods by introducing more advanced cardinality estimation approaches. Adding comparisons with data-driven cardinality estimation methods or experiments against paradigmatic methods like QueryFormer would enhance the analysis.
W3. The experimental workload in this paper lacks clarification regarding query redundancy. It should include comparative experiments under different workloads. Additionally, experiments on cardinality estimation with lower query redundancy should be added to provide a more comprehensive evaluation.

**Questions:**

Q1. The addition of mean absolute error (MAE) and relative prediction error (RPE) would allow for a more comprehensive evaluation of the accuracy of different cardinality estimators.
Q2. It should include a broader variety and greater number of baseline methods by introducing more advanced cardinality estimation approaches. Adding comparisons with data-driven cardinality estimation methods or experiments against paradigmatic methods like QueryFormer would enhance the analysis.
Q3. It should include comparative experiments under different workloads. Additionally, experiments on cardinality estimation with lower query redundancy should be added to provide a more comprehensive evaluation.
Q4. Could you clarify what specific models are used at each level of cardinality estimation in this paper? This detail is not adequately explained in the manuscript.

---

> ### Author Response · Authors · 2024-11-19
>
> W1+Q1: We added Absolute Error and Relative Error in Table 2 and Table 3.
> We added absolute error and relative error to the paper (see metrics, at start of Section 4, and also experiment Tables 2 and 3) -- thanks a ton! Indeed, this makes the experiments more thorough.
>
> W2+Q2: We are adding DeepDB instead of QueryFormer (see results on top), as it is data-driven approach and complimentary to MSCN. Nonetheless, as discussed in Motivation in main comment to all reviewers (above), we think it is orthogonal to our method to compare it with baselines as our method **needs a baseline, at the root node, to fall back on, for novel queries**.
>
> W3+Q3: We improved the wording in the paper. Real workloads (e.g., per Redshift paper) have repetitions in query patterns. Nonetheless, your proposed experiments makes absolute sense.
>
> We add more details in **Datasets** of the experiments section and ablation studies. Our difference from the Cardbench dataset is that we modified the workload generator to allow the query predicates have more constants to choose from. All multijoin datasets we created fixed the constant sample size at 10, resulting in a repetition rate near 90% (the number reported in the Redshift paper). Additionally, we have conducted a small sweep on the sample size (ie. [1, 3, 10]) for multijoin-stackoverflow. See Fig 4 in the ablation studies. It shows that accuracy generally improves when we introduce more repetition rate. And even when the repetition rate is low, our history-based learner still shows promising results.
>
> Q4: In the ablation studies section, we showed results on different models. We found that Gradient-Boosted Decision Trees (GBDT) are consistently strong and used this type of model at each level of the hierachy.

---

### Official Review · Reviewer_5CiV · 2024-11-02

**Soundness:** 1
**Presentation:** 1
**Contribution:** 1
**Rating:** 3
**Confidence:** 4

**Summary:**

The authors propose a supervised cardinality estimation method that enhances accuracy for workloads with repetitive queries (i.e., queries repeated in templates with only the constant parameters changed) by leveraging hierarchical, online localized models. This approach transforms each SQL query into a Directed Acyclic Graph (DAG) representation, groups isomorphic DAGs into the same partition, and trains an online model for each partition as queries are executed. Grouping is conducted hierarchically at multiple levels of granularity, ranging from fine-grained (i.e., queries varying only in constant terms are placed in the same partition) to coarse-grained (e.g., queries varying only in constants, operator types, and column names are placed in the same partition). During runtime, given a query, the method begins with the fine-grained model and moves to coarser-grained models until it finds a confident model for the query. If no suitable model exists, it defaults to a learned or traditional model. Using the CardBench benchmark, the authors demonstrate that this method yields more accurate cardinality estimates compared to competitors such as PostgreSQL and MSCN.

**Strengths:**

S1. The authors tackle an important issue in analytical databases: cardinality estimation for repetitive queries.

S2. The authors propose a cardinality estimator, which leverages hierarchical localized models that are trained online.

**Weaknesses:**

W1. The Experiments section needs to be improved.
- The authors should compare their method with state-of-the-art (SOTA) data-driven cardinality estimation methods such as [1]. Currently, the comparison is limited to MSCN, a query-driven method proposed in 2019 [2], and PostgreSQL, a traditional estimator.
- The “Imperfect Admission Experiments” in Section 4 seems unfair, as the q-error percentiles of each method are reported on different query subsets: PostgreSQL is evaluated on the entire query set, MSCN on a subset excluding disjunctions, while the proposed method seems to be evaluated on a subset of simple queries (e.g., repetitive queries varying only in constant terms).
- The authors should report end-to-end time (including both planning and query execution time), as shown in [1,3], along with its breakdown for further clarity.
- Reporting training time and model size would provide further insights into the method’s practical feasibility.
- The authors should clarify why the q-error is higher in certain intervals with larger training sample sizes in Figure 4.
- The experimental setup requires a more thorough explanation, particularly regarding how query repetitiveness was generated and its extent.


W2. Motivation is inadequate.
- The authors should explain why existing learned estimators struggle with repetitive workloads to highlight the necessity for their proposed method.
- The authors should clarify why a hierarchical model structure is effective in improving the accuracy of cardinality estimation.

W3. The presentation needs to be improved.
- The process of generating a DAG from a query needs further explanation. If the DAG refers to a query plan, the authors should specify which query plan is used, as multiple plans can exist for a single query.
- There are some undefined terms, such as d_{\psi} in Section 2.
- There are inconsistent notations throughout the paper. For instance, “query graph” and “query plan” are used interchangeably.
- Numerous typos appear throughout the paper, such as “geoping” and “hases” in Section 5.

W4. There are some misstatements regarding existing work
- The authors seem to overstate the limitations of existing methods. They claim that “NN-based estimators perform well if they are trained with large amounts of query samples,” which is true specifically for query-driven learned estimators, not all NN-based methods.
- The authors state that “50% of the real world clusters have more than 90% queries repeated in templates (only changing the constant parameters),” citing [4]. However, according to [4], the correct value is 80%, not 90%.

[1] Kim, Kyoungmin, et al. "Asm: Harmonizing autoregressive model, sampling, and multi-dimensional statistics merging for cardinality estimation." Proceedings of the ACM on Management of Data 2.1 (2024): 1-27.
[2] Kipf, Andreas, et al. “Learned cardinalities: Estimating correlated joins with deep learning.” In Biennial Conference on Innovative Data Systems Research, 2019.
[3] Wu, Ziniu, et al. "FactorJoin: a new cardinality estimation framework for join queries." Proceedings of the ACM on Management of Data 1.1 (2023): 1-27.
[4] van Renen, Alexander, et al. "Why tpc is not enough: An analysis of the amazon redshift fleet." Proceedings of the VLDB Endowment 17.11 (2024): 3694-3706.

**Questions:**

Please refer to W1-W4.

---

> ### Author Response · Authors · 2024-11-19
> **Part 1 / 2**
>
> ## W1
>
> * We are adding baseline based on reviews. We went with data-driven-approach DeepDB. However, we added [1] to the related work section.
>
> * Unfairness of eval: Due to this, we re-designed our experimental setup, as discussed in the common Comment above.
>
> * We plan to add P-error (we already started working on it!) which gives some quantification of the improvement of the end-to-end time.
>
> * The time for our methods to run: Pop-out features from query graph, hash it, lookup model and run it, is fast -- each feature pop + graph hash averages 2.3 milliseconds. Each inference step involves 3 hashes (one per templatization strategy) [i.e., total hash time, per graph = 7 milliseconds on avg]. The inference time is negligable (e.g., microseconds) and the cost to train is model-dependent. For example, we can train 15000 linear regression models in under 5 seconds -- thanks to fast implementation of numpy pinv. For decision forests, we use xgboost -- it takes around half-a-second to do training (slow!). However, we will be moving to TensorFlow Decision Forests (TFDF) as we heard they are orders of magnitude faster. **We will have the exact time numbers onto the main paper**, by the paper camera ready (if paper gets accepted), as we would like to invoke the fingerprint in C++ to yield further advantages, and also test the speed of TFDF. This speed makes us name our method as "online".
>
> * Fig4: now moved to appendix and became Fig 5. We improved it. It used to be: Q-error percentile (at y-axis) measured **at** history size **equal** to x-axis. That plot was very noisy. For instance, for large history size (e.g., >100), there may be one (or a couple of) template(s) with that many and therefore the Q-error estimate would come from one (or a couple of) observation(s). Now it became a cumulative plot: it measures q-error for the amount of history up-to the x-axis value (e.g., Curve(10) = Q-error when there are <= 10 examples). Curve looks less noisy. However, for coarser templatization, it does not clearly converge with more observations (due to multimodality e.g. query predicate on different columns can group to same bucket) -- Thanks for the constructive feedback.
>
> * We now offer more details in Experiment **Datasets** section and **Repetition Rate** section in Ablation Studies. In short, we modified the workload generator to allow the query predicates have more constants to choose from. In this way, the repetition rate is close to 90% across different datasets (ie. the repetition rate number reported in [4]). We also show in the ablation studies section that the accuracy generally improve when we introduce more repetition.
>
> ## W2
> * We added it in the main response.
>
> * Importantly, our online learning setup avoids the need for a complex training data pipeline, which is the biggest barrier for practical ML-based cardinality estimation techniques. While NN models, whether workload-driven or data-driven, often perform well, they can still produce significant errors (q-errors > 3), which can have severe consequences. By focusing on frequently occurring query patterns, we can significantly reduce their q-error to below 2.

---

> ### Author Response · Authors · 2024-11-24
> **Part 2 / 2**
>
> ## W3
> * We use a logical query plan to create the query graph (or DAG). We used CardBench to generate the logical query plans, which does not apply any transformations to the query plan, but our methods would work with any optimizer.
> For the same exact SQL string we will produce the same logical query plan.  A join (B join C) and (A join B) join C will be result in two different logical query plans, and hence will not produce the same DAG. However, your review (and another reviewer) is making us think about canonicalizing the graph (e.g., making super node that absorbs all join nodes). Nonetheless, as-is, our method has some false negatives (i.e., one equivalent query might be partitioned onto multiple graphs), and this could be hurting the performance. In the final version, we will either fix this in code **or** mention it in the paper (we will try to go for fixing the implementation, first).
>
> * We added text (in blue): where $d_{\psi} \in \mathbb{Z}_+$ is dimensionality of extracted feature -- thank you.
>
> * We now consistently use "query graph" -- thank you! We feel this is better aligned with ICLR audience. However, if this paper gets rejected, we will consider renaming to "query plan" then submit to database venue (especially that DB folks we talk to, are all thrilled by this work).
>
> * We fixed a lot of typos. We will give the paper more thorough reads, before we post it anywhere (e.g., camera ready or arxiv). Thank you!!
>
> ## W4
>
> * We now compare with data-driven neural methods. It seems that we outperform in one case and they outperform in another (for 2 datasets). However, training takes hours (using code open-sourced by authors of DeepDB).
>
> * We acknowledge that [4] reports that "50% of database clusters have 80% of queries as 1-to-1 repetitions." However, this statistic refers to *exact query repetition*, where the entire query string is identical to a previous query. In our work, we focus on a broader notion of query repetition, leveraging query templates. As highlighted in Figure 5(c) of [4], 50% of clusters exhibit more than 90% query repetition within templates over a one-week period. This indicates a high degree of structural similarity among queries, even if they differ in parameter values.

---

### Official Review · Reviewer_3TiU · 2024-11-03

**Soundness:** 3
**Presentation:** 2
**Contribution:** 2
**Rating:** 6
**Confidence:** 5

**Summary:**

This paper proposes a workload-driven approach for cardinality estimation aiming at workloads containig repetative queries. The proposal utilizes multiple templatizer to derive hierarchical cardinality estimation models, taking a query plan tree as input and obtaining data with different granularity. It employs general predictors like PostgreSQL to perform cardinality estimation at the lowest level.

**Strengths:**

- S1: The evaluation experiments used six types of workloads, including up to five-table join queries, and achieved a higher Q-error than MSCN (CIDR2019). It was confirmed that the fine-grained model, H1, achieved the fastest convergence in training and the highest accuracy.

- S2: Since machine learning-based methods often underperform in high-percentile cases, the fallback mechanism is beneficial in practice. As shown in Table 4, the deeper hierarchical fallback mechanism achieves a higher Q-error.

- S3: As a query-driven approach, the featurizer’s ability to characterize predicates is useful.

**Weaknesses:**

- W1: In addition to Q-error, P-error should be also evaluated, which is becoming a standard metric.

- W2: As shown in Table 3, H1 demonstrates the highest performance, so a structure of H1 -> PostgreSQL seems optimal, making the multiple hierarchy setup (in Equation 15) unnecessary. The authors should clarify the benefit of using multiple hierarchy levels.

- W3: Although the proposal adopts a workload-driven approach, recent trends favor data-driven or hybrid approaches. Notably, data-driven approaches have the advantage of robustness for unknown queries. Combining a workload-driven approach with data-driven methods could enhance accuracy in cases where prediction errors are large. While a workload-driven approach has the advantage of faster inference time, it is not obvious workload-driven approach alone is useful.

- W4: In Section 2.5, the statement "All graphs whose templates are isomorphic share the same model" is based on graph isomorphism as defined in Definition 1, relying on edge information only. The templatizer, H1, thus does not use graph attribute information. However, Section 3.1 states, "Hence, query graphs found in the same H1 template differ only by the constant values," which appears contradictory.

- W5: Section 3.3 includes a calculation for the hash size, such as s(#T1), but according to Equation 6, the hash length is fixed at h, so the case distinction in Equation 10 does not seem valid.

- W6: Although Cardbench is used as the evaluation benchmark, it is proposed in an arXiv paper and is not yet a standard benchmark. It would be better to use the widely accepted join order benchmark or explain the strong justification for using Cardbench.

**Questions:**

- It is unclear how the plan tree is constructed. For example, is it correct to interpret that (A ? B) ? C and A ? (B ? C) are non-isomorphic plans?

---

> ### Author Response · Authors · 2024-11-19
>
> * **W1**: We looked into P-error. It quantifies the impact of cardinality estimation on the real metric (cost of query execution). However, its implementation requires some time. It seems we have to invoke Postgres query-planner twice, once using cardinality hints (of subgraphs) from a model, and once again feeding the ground-truth, giving two plans $P(C^E)$ and $P(C^T)$. We should then run Postgres query plan cost-estimator twice, once on $(P(C^E),  C^T)$ and another on $(P(C^T),  C^T)$ and compute the ratio of the two -- crucially feeding ground-truth cardinalities $C^T$ to both cost-estimator invocations. This makes sense, as it shows the bottom-line metric, i.e., quantifying the slowness introduced by inaccurate cardinality estimates, as compared to the optimal [achievable by the query-planner].
>
> Implementing P-error implies we have to integrate with the (1) Postgres query-planner and plan cost-estimator and (2) annotate cardinalities at subqueries (as you mention, this can be retrieved during execution, and our online learner can absorb that info). Alternatively, we can integrate with https://github.com/Nathaniel-Han/End-to-End-CardEst-Benchmark which we found upon searching per your review. We will do our best to implement this by the camera read deadline, as it can greatly improve our paper.
>
> * **W2**: While $H_1$ are indeed the best hierarchy, it only captures 45\%-75\% of the queries [as many queries differ only in their constant parameter value] and $H_2$, $H_3$, ..., are designed to capture the remaining ones [e.g., query on same table though different column predicate], as much as possible, though we cannot capture more than 26% with any $H_i$ on the datasets.
>
> In addition to the experiments comparing  $H_1 \rightarrow H_2 \rightarrow H_3 \rightarrow Postgres $ VS $H_2 \rightarrow H_3 \rightarrow P$ VS  $H_3 \rightarrow P$, we have additional experiments now also comparing  $H_1 \rightarrow P$, $H_1 \rightarrow H_2 \rightarrow P$. Generally, $H_1 \rightarrow H_2 \rightarrow H_3 \rightarrow P$ seems to be the strongest. See updated Table 2.
>
> * **W3**: Although the proposal adopts a workload-driven approach, recent trends favor data-driven or hybrid approaches. Notably, data-driven approaches have the advantage of robustness for unknown queries. Combining a workload-driven approach with data-driven methods could enhance accuracy in cases where prediction errors are large. While a workload-driven approach has the advantage of faster inference time, it is not obvious workload-driven approach alone is useful.
>
> we understand that our approach does not cover unseen queries. Our goal is to improve the error rate of frequently run queries (which is about ~80% of workloads).  That is why we fall back to a general model as the default. In our experiments we used postgres, but this model can be any of the learned cardinality models as well (query- or data-driven). Specifically, one possible future work is to replace the *root model* by a data-learned model (e.g., GNN), which we elude to, in the *fifth-line in the Conclusion*. This could combine the advantages of both worlds (no restriction on input query, yet ability to specialize per workload).
>
> * **W4**: We sincerely apologize. Our Definition also associates node features. We had a typo in the definition $f^{(v)} = z^{(\pi_v)}$ that is now corrected to $f^{(v)} = f^{(\pi_v)}$ [we somehow missed that earlier, during a rename exercise $z \rightarrow f$]. We checked: the Appendix hashing algorithm uses the correct symbol $f$.
>
> * **W5**: We sincerely apologize (again!). We (mistakenly) omitted notation $s(.)$ in the submitted version. We fixed it now by adding:
>
> where the history size $s(Ti)$ equals the number of observations that hash to ${T_i}$, \textit{i.e.}, the height of matrices $\mathbf{X}_i^{[T_i]}$ and $\mathbf{Y}_i^{[T_i]}$.
>
> (it is marked in blue)
>
> * **W6**: We use CardBench due to practical advantages. First, they open-source their query generator (with instructions on how to re-generate the data). We modified their query generator to repeat patterns (but we remove duplicate queries) to represent common workloads where queries repeat.
>
> JOB benchmark uses IMDB, which has a license not compatible with our corporate policies. But our workload generator follows similar patterns as JOB.
>
> Q: It is unclear how the plan tree is constructed.
>
> Excellent point. We have been relying on the query parser of CardBench to convert SQL statement into query graph. While CardBench is consistent (graph is determined by order of join conditions), we may be splitting identical graphs onto different buckets. We think the graph can be canonicalized (e.g., super-node can absorb all join nodes). Worst-comes-worse, it is OK to have equivalent query into different buckets [potentially this can cause degradation of metrics]. We will do our best to have this aspect reflected into the paper and into our final implementation, by camera ready, if our paper is accepted.

---

> > ### Comment · Reviewer_3TiU · 2024-11-21
> >
> > Thank you for the responses. All of them are reasonable. I raised the score by 1 point.
> > I understand that the P-error-based evaluation requires a lot of effort.

---

### Author Response · Authors · 2024-11-19
**Thank you for your reviews!**

Dear reviewers,


We took our time to craft the response, as your feedback pushed us to re-do our experimental setup, fix typos in paper, and discuss some more related work. We uploaded the new rebuttal PDF. We highlight significant changes in blue.

Our response here addresses common concerns. Additionally, we respond to each reviewer, often referencing this very response.

Given your feedbacks, we have done the following:

## Re-design Experiments

1. We run another data-driven baseline: DeepDB[1]. We run the code of authors of DeepDB (we only transform the input format) and show the comparisons on two datasets. DeepDB outperforms our method on **multijoin-accidents** dataset but underperforms on **multijoin-stackoverflow** dataset (numbers are at the end of this comment).
2. Remove "imperfect admission" experiments since reviewers eluded that they are not fair (each method potentially evaluated on different set of queries). Instead, we evaluate all methods on the intersection of queries they are all able to process in Table 3 of the revised PDF. This shows that: when query pattern is known, our methods outperform the baselines. This then motivates the hierarchy experiments (use our models for repetive queries) -- Table 2 of the revised PDF. In most practical scenarios, only Postgres estimator can always compute (as it can handle arbitrary queries and does not require training), that's why we use it at the root in our experiments.
3. We are now reporting two additional metrics (Absolute error and Relative error) -- Unfortunately, integration with P-Error requires more work, because it needs to invoke cardinality estimator at each subquery, and additionally invoke Postgres query-planner and cost-estimator during eval. We are working on it, and we plan to have the numbers in our final version.
4. We add the ablation study on different repetition in Fig 4 of the revised PDF.
5. We replaced experiment figure (now Fig #5 in Appendix) with ``cumulative plot'' i.e. the errors when using this-many-prior-observations-or-less (versus the old one: using this-many-observations) as the old one was noisy: at high example count, the number would come from one (or a couple) of predictions.

## Motivation

We want to emphasize that our contribution learns from queries (online) by grouping similar ones. **Our work must sit on-top** of another cardinality estimator (may it be deep net or heuristic-based). The familiar queries can be intercepted by our hierarchy and the unseen queries can fall back onto more-general estimator. Since Postgres can always process any query, and that it gives decent results [given no training!], we use it as a default fall-back. Since our method does **not** stand on its own [needs a fallback for novel templates], it is somewhat orthogonal to compare it against many baselines.

## DeepDB Comparison

We report Q-Errors (at 50th and 90th percentile) and also median absolute and relative errors, for two multijoin datasets. We train on 100,000 sample rows per table -- training takes several hours, per dataset. We are queueing runs for the remainder of the datasets, and we should have all numbers reported by camera ready (if paper gets accepted).


|               |                | Postgres | MSCN+  | DeepDB    | Ours      |
|---------------|----------------|----------|--------|-----------|-----------|
| multijoin-stackoverflow | Q-error @50    | 3.8      | 2.0    | 3.9       | **1.05**  |
|               | Q-error @90    | 149      | 10.99  | 1.9e4     | **2.19**  |
|               | Relative Error | 0.74     | 0.50   | 0.74      | **0.05**  |
|               | Absolute Error | 4.2e4    | 9.7e3  | 2.7e5     | **30.3**  |
| multijoin-accidents     | Q-Error @50    | 1.73     | 2.2    | **1.02**  | 1.09      |
|               | Q-error @90    | 11.04    | 8.31   | **1.17**  | 3.31      |
|               | Relative Error | 0.42     | 0.54   | **0.02**  | 0.08      |
|               | Absolute Error | 6.0e7    | 3.1e7  | **1.4e6** | 4.0e6     |


[1] Hilprecht, Benjamin, et al. "DeepDB: Learn from Data, not from Queries!", VLDB 2020.

---

### Meta-Review · Area_Chair_rrKA · 2024-12-20

**Metareview:**

This paper proposes a workload-driven approach for cardinality estimation for workloads containing repetitive queries. The reviewers agree the problem is important and that the hierarchical cardinality estimation methods are effective. However, there are also concerns on the motivation, related work, experiments, and presentation. While some of the comments were addressed during the discussion period, others seem to be only partially addressed with ongoing experiments. Overall, I think the paper can be improved and should go through another round of reviewing.

**Additional Comments On Reviewer Discussion:**

I will focus on the discussion on experiments. First, the integration with P-Error was suggested by reviewer 3TiU and was later considered to require too much effort, but the authors also mention to reviewer 5CiV that they are performing this experiment. Second, the comparison with DeepDB seems critical as the current baselines are considered relatively outdated (MSCN and MSCN+). While the authors explain DeepDB is orthogonal to the proposed method, both reviewers 5CiV and 2Nam do not seem convinced. Also saying that some results will be available if the paper is accepted is not convincing either. Hence, the discussion on experiments seems largely inconclusive to me.

---

### Decision · Program_Chairs · 2025-01-22

Reject